# Antibacterial Activity of Bioactive Compounds Extracted from the Egyptian Untapped Green Alga *Rhizoclonium hieroglyphicum*

**Hanaa H. Morsi [1], Sabha M. El-Sabbagh [1], Ahlam A. Mehesen [2], Ahmed D. Mohamed [3], Maha Al-Harbi [4], Amr Elkelish [5,6], Mostafa M. El-Sheekh [7] and Abdullah A. Saber [8,***

1. Botany and Microbiology Department, Faculty of Science, Menoufia University, Shebin El-Kom 32511, Egypt
2. Algae Unit, Soil, Water and Environment Research Institute, Agriculture Research Station, Sakha 33717, Kafr El-Sheikh, Egypt
3. Drinking Water and Sanitation Company, Sidi Salem 33743, Kafr El-Sheikh, Egypt
4. Department of Biology, College of Science, Princess Nourah bint Abdulrahman University, P.O. Box 84428, Riyadh 11671, Saudi Arabia
5. Biology Department, College of Science, Imam Mohammad ibn Saud Islamic University (IMSIU), P.O. Box 90950, Riyadh 11623, Saudi Arabia
6. Botany and Microbiology Department, Faculty of Science, Suez Canal University, Ismailia 41522, Egypt
7. Botany Department, Faculty of Science, Tanta University, Tanta 31527, Egypt
8. Botany Department, Faculty of Science, Ain Shams University, Abbassia Square, Cairo 11566, Egypt
* Correspondence: abdullah_elattar@sci.asu.edu.eg; Tel.: +20-10-0256-2239

**Abstract:** Finding alternative powerful antibacterial drugs of natural origins is, today, a crucial pre-requisite due to the resistance of some bacterial strains to commercial and widely-used medications. Algae are characterized by their bioactive constituents and have a wide spectrum of biotechnological aspects, particularly antibacterial implications. During this study, four concentrations (5, 10, 20, and 40 mg mL$^{-1}$) of the Egyptian untapped green microalga *Rhizoclonium hieroglyphicum* (Chlorophyta) were prepared using the polar solvents ethanol, methanol, and acetone. The antibacterial activity of the above-mentioned extracts was assessed, using the agar disc diffusion technique against three pathogenic bacteria, *Staphylococcus aureus* ATCC 6538, *Escherichia coli* ATCC 8739, and *Pseudomonas aeruginosa* ATCC 9027, which was compared to standard antibiotics. The minimal inhibitory concentrations (MICs) were also assessed and determined using a broth dilution assay. Our findings revealed that the *R. hieroglyphicum* ethanolic extract exhibited the most potent antibacterial effect and its MICs values were 0.533, 2.25, and 5.34 mg mL$^{-1}$ against *P. aeruginosa*, *E. coli*, and *S. aureus*, respectively. A gas chromatography–mass spectrometry (GC–MS) approach to the crude *R. hieroglyphicum* ethanolic extract uncovered 30 different bioactive constituents, mainly including long-chain polyunsaturated and saturated fatty acids such as myristic (C14:0), palmitic (C16:0), stearic (C18:0), α-linolenic (C18:3; ω–3), and oleic (C18:1, ω–9) acids, which synergistically make this potent antibacterial action. The mechanism of action of these fatty acids was also discussed. Conclusively, *R. hieroglyphicum* could be a good candidate for the production and development of promising antibacterial agents.

**Keywords:** *Rhizoclonium hieroglyphicum*; pathogenic bacteria; algae-derived antibiotics; antibacterial metabolites; GC-MS; long-chain fatty acids

## 1. Introduction

Taxonomically, the green filamentous alga *Rhizoclonium* Kützing is placed within the order Cladophorales (Chlorophyta), differs from the closest genus *Cladophora* in its distinctive phylogenetic position, and lacks branches in most of its species [1–3]. Ecologically, it is most often found in freshwater environments, with a few species that can tolerate medium-to-high salinity conditions [4]. *R. hieroglyphicum* (C. Agardh) Kützing is the most widely distributed species of the genus *Rhizoclonium* and it typically prefers freshwater

habitats rich in nutrients. One of the most intriguing characteristics of *R. hieroglyphicum* is its richness of cell wall polysaccharides; therefore, it is utilized as a biopolymer in several applications such as thickening agents in the food industry and the manufacture of pharmaceuticals, but its antimicrobial applications are still an underestimated topic [5].

Algae, including cyanoprokaryotes, eukaryotic microalgae, and seaweeds, are a natural source of high-value bioactive constituents characterized by their immense biotechnological applications [6–11]. Given their potential antimicrobial effects against several pathogenic bacteria and fungi, they are currently utilized in different pharmaceutical industry trends and for medication improvement [9,12,13]. The algal extracts and their highly effective bioactive fractioned compounds have already been documented to display a wide spectrum of antimicrobial actions towards several pathogenic bacteria, fungi, and viruses [6,14–16]. For instance, Ismail and co-workers [17] investigated the antibacterial action of the marine green macroalga *Ulva rigida* against *Staphylococcus aureus* ATCC 25923 and *Enterococcus faecalis* ATCC 29212, identifying 16 different bioactive fractions responsible for these antibacterial effects. Fatty acids (FAs), primarily oleic, linoleic, palmitic, and stearic acids, were among the major constituents of *U. rigida*. They also reported that the MIC values ranged from 10 to 250 μg mL$^{-1}$. With regard to the mechanism of the antibacterial action of *U. rigida*, they proposed that palmitic acid may work in concert with oleic acid to provide antibacterial inhibition. Additionally, they suggested that stearic acid appeared to play a part in the antibacterial activity that was noticed, particularly against *S. aureus* ATCC 25923. The cosmopolitan green microalga *Scenedesmus obliquus* (currently known taxonomically as *Tetradesmus obliquus*) and *Haematococcus pluvialis* have also gained the interest of several researchers worldwide. Their antibacterial characterizations towards methicillin-resistant *S. aureus* and *E. coli* have also been investigated [18,19]. These antibacterial actions were mainly linked to their immense polyunsaturated FAs. The marine diatom species *Phaeodactylum tricornutum* has also been investigated and its antibacterial action against methicillin-resistant *S. aureus* has been documented to be due to its polyunsaturated eicosapentaenoic and hexadecatrienoic FAs, as well as its monounsaturated palmitoleic FAs [20]. All the previous studies have extensively confirmed that saturated and unsaturated FAs, with medium and long chains, play key efficient roles against both Gram-positive and Gram-negative bacteria. Despite the extensive antimicrobial studies on a wide range of cyanobacterial and algal species, our knowledge of the antimicrobial potential of other algal taxa is still incomplete.

Indeed, the altered therapy of incurable diseases with medications has specific limitations due to ongoing changes in the microbial infection mechanisms and adverse complications that they cause. These challenges demand extensive research to find out novel antibacterial compounds for the development of new drugs, in order to improve their pharmacokinetic characteristics [21]. The exploration of novel and powerful antibacterial drugs from algae is highly promising, currently required to beat multi-drug resistant bacteria, and also to control the increased risk of incurable diseases caused by these pathogenic microorganisms [9,22]. In general, our knowledge about antimicrobial algal extracts is still underestimated and more in-depth research is needed in this respect of research.

Although the filamentous green alga *Rhizoclonium hieroglyphicum* (Cladophorales, Chlorophyta) is widely distributed worldwide, our knowledge about its antimicrobial characterization is still poor. The present research aimed to assess the In vitro antibacterial implications of three different extracts (ethanol, methanol, and acetone) of *R. hieroglyphicum*, mostly untapped in Egyptian inland waters, against three pathogenic bacterial strains, namely *Staphylococcus aureus*, *Escherichia coli*, and *Pseudomonas aeruginosa*. As it is the most potent one, an analysis of the crude *R. hieroglyphicum* ethanolic extract was also conducted for characterizing the bioactive antibacterial substances using a gas chromatography–mass spectrometry (GC–MS) approach.

## 2. Materials and Methods

### 2.1. Algal Origin and Culture Conditions

*Rhizoclonium hieroglyphicum* was isolated from the freshwater stream Ruwaynah, Bahr Nashart, Kafr El-Sheik Governorate, Egypt. The purified strain was grown in BG11 liquid medium at 24 ± 1 °C and a 16:8 h L/D photocycle with 20 W cool white fluorescent lamps, which provided a light intensity of approximately 500 Lux [23].

### 2.2. Bacterial Strains

The authentic bacterial strains *Pseudomonas aeruginosa* ATCC 9027, *Escherichia coli* ATCC 8739, and *Staphylococcus aureus* ATCC 6538 were taken from the Botany and Microbiology Department, Faculty of Science, Menoufia University, Shebein El-Kom, Egypt.

### 2.3. Preparation of the Algal Extracts

The *R. hieroglyphicum* specimens were grown until they reached the late exponential growth phase, when they then were collected using centrifugation at 5000 rpm for 10 min. The algal pellets were gathered, dried, and weighed. In total, 0.5 g of the algal pellet was immersed independently in ethanol, methanol, and acetone (1:10 $w/v$) at 4 °C until it was fully extracted. The supernatants were obtained after centrifugation at 10,000 rpm for 10 min [24]. The dried aggregate produced from each extraction process was re-dissolved again in the same solvent to obtain a final concentration of 40 mg mL$^{-1}$. The final extracts were kept at 4 °C until further antibacterial assays were used [25].

### 2.4. Antibacterial Assay of the Algal Extracts

Four different concentrations (5, 10, 20, and 40 mg mL$^{-1}$) of each algal extract were used to assess the antibacterial activity using an agar disc diffusion assay [26]. Briefly, nutrient agar medium was sterilized using autoclaving at 121 °C and 1.5 lbs of pressure for 15 min. A total of 100 μL of bacterial cells were inoculated onto the Petri dishes, and 6 mm sterilized filter paper discs, loaded with 50 μL of each algal crude extract and subject to air drying, were placed onto the bacteria-inoculated Petri dishes. Each solvent was also tested to confirm the antibacterial algal effects. The plates were incubated at 37 °C for 24 h. At the end of the incubation period, the zones of inhibition around the paper circles were determined in millimeters (mm). The standard antibiotics (ampicillin, amoxicillin, cefadroxil, doxycycline, cefoxitin, ofloxacin, and vancomycin) were used as positive controls for comparison.

### 2.5. Determination of the Minimal Inhibitory Concentrations (MICs) of the Algal Extracts

The MICs of the algal extracts, exhibiting considerable antibacterial activity, were determined using a dilution technique [27]. The algal extracts were gradually diluted in test tubes in range of 0.225 until they reached 40 mg mL$^{-1}$. In total, 0.1 mL of each bacterial suspension, containing $2 \times 10^6$ CFU mL$^{-1}$, was added into each tube. The tubes were incubated with the standard growth conditions (37 °C for 24 h). The control tubes were without any algal extract. The lowest concentration at which there was not any visible growth was considered as the MIC.

### 2.6. Identification of the Bioactive Compounds of the R. hieroglyphicum Ethanolic Extract

The bioactive constituents in the *R. hieroglyphicum* ethanolic extract, as it was the most potent antibacterial solvent in this study, were analyzed using gas chromatography–mass spectrometry (GC–MS) using a Perkin Elmer Elite-5MS capillary column (30 m × 0.25 mm ID, 0.25 μm film thickness) in a 7890B gas chromatograph system (Agilent Technologies Co., Santa Clara, CA, USA), coupled with a 5977A mass selective detector. Helium was the carrier gas, with a 1.8 mL min$^{-1}$ flow rate. The algal extract injected was 1.0 μL. The fractioned bioactive constituents were unraveled by comparing their mass spectra with those in the standard databases [28].

*2.7. Statistical Analysis*

All the experiments were carried out in triplicate and the data were statistically analyzed with the SPSS package (v. 25, SPSS Inc., Chigaco, IL, USA). All the readings were expressed as mean ± standard deviations (SD). A two-way ANOVA test was applied to examine the effect of the solvent type and different concentrations of *R. hieroglyphicum* on the tested pathogenic bacterial strains. Tukey's tests were performed to compare the statistically significant differences among the means at $p < 0.05$.

## 3. Results

*3.1. Antibacterial Activity of the Different R. hieroglyphicum Extracts*

The antibacterial activities of the different *R. hieroglyphicum* extracts against the three bacterial strains, *E. coli*, *P. aeruginosa*, and *S. aureus*, are represented in Table 1. The averages of the inhibition zones of the ethanolic extract at a concentration of 40 mg mL$^{-1}$ were 15.2, 18.2, and 13.2 mm, respectively; 13.3, 16.3, and 11.3 mm at a concentration of 20 mg mL$^{-1}$; and 11.3, 14.3, and 8.28 mm at 10 mg mL$^{-1}$, respectively. Inhibition zones were only recorded for *E. coli* and *P. aeruginosa* at a concentration of 5 mg mL$^{-1}$, with average values of 7.21 and 10.2 mm, respectively. With regard to the methanolic extract, the averages of the inhibition zones at 40 mg mL$^{-1}$ were 7.12, 10.2, and 15.1 mm against *E. coli*, *P. aeruginosa*, and *S. aureus*, respectively. At a concentration of 20 mg mL$^{-1}$, the inhibition zones were 8.12 for *P. aeruginosa* and 13.2 for *S. aureus*, while there was no effect against *E. coli*. Interestingly, *S. aureus* was inhibited at concentrations of 10 and 5 mg mL$^{-1}$ (the averages of the inhibition zones were 10.3 and 9.17 mm, respectively). With respect to the acetone extracts, they had remarkable antibacterial activities against *E. coli*, with inhibition zones of 13.2, 11.3, and 8.32 mm at the concentrations of 40, 20, and 10 mg mL$^{-1}$, respectively, but no response at 5 mg mL$^{-1}$ (Table 1). The inhibition zones of the *R. hieroglyphicum* acetone extracts against *P. aeruginosa* scored 14.3, 12.2, 10.3, and 8.15 mm at concentrations of 40, 20, 10, and 5 mg mL$^{-1}$, respectively. Lastly, *S. aureus* showed inhibition zones of 13.25 mm at 40 mg mL$^{-1}$, 11.3 mm at 20 mg mL$^{-1}$, 9.20 mm at 10 mg mL$^{-1}$, and 7.15 mm at 5 mg mL$^{-1}$.

**Table 1.** In vitro antibacterial activities of the different *Rhizoclonium hieroglyphicum* extracts against the tested pathogenic bacterial strains using agar disc diffusion assay (data represent the final thickness of the inhibition zones in mm after comparing them with the solvents used).

| Algal Extracts | Conc. (mg mL$^{-1}$) | *E. coli* | *P. aeruginosa* | *S. aureus* |
|---|---|---|---|---|
| ethanolic extract | 40 | 15.2 ± 0.18 [a] | 18.2 ± 0.22 [a] | 13.2 ± 0.17 [a] |
| | 20 | 13.3 ± 0.17 [b] | 16.3 ± 0.25 [b] | 11.3 ± 0.23 [a] |
| | 10 | 11.3 ± 0.26 [b] | 14.3 ± 0.27 [c] | 8.28 ± 0.39 [b] |
| | 5 | 7.21 ± 0.17 [c] | 10.2 ± 0.09 [d] | – |
| methanolic extract | 40 | 7.12 ± 0.05 [a] | 10.2 ± 0.08 [a] | 15.1 ± 0.06 [a] |
| | 20 | – | 8.12 ± 0.04 [b] | 13.2 ± 0.13 [a] |
| | 10 | – | – | 10.3 ± 0.29 [b] |
| | 5 | – | – | 9.2 ± 0.12 [c] |
| acetone extract | 40 | 13.2 ± 0.19 [a] | 14.3 ± 0.19 [a] | 13.3 ± 0.18 [a] |
| | 20 | 11.3 ± 0.20 [b] | 12.2 ± 0.11 [b] | 11.3 ± 0.18 [a] |
| | 10 | 8.32 ± 0.17 [b] | 10.3 ± 0.26 [c] | 9.20 ± 0.18 [b] |
| | 5 | – | 8.15 ± 0.11 [d] | 7.15 ± 0.11 [c] |

Notes: Data are represented as means ± SD (*n* = 3). Means in the same column with different letters are significantly different at $p < 0.001$.

*3.2. Antibacterial Action of the Positive Standard Antibiotics*

As shown in Table 2, doxycycline at 30 μg and ofloxacin at 5 μg only exhibited antibacterial activities against the three tested bacterial strains, with the averages of their inhibition zones being 15.2 and 16.4 mm against *E. coli*, and 9.11 and 6.12 mm against *P. aeruginosa*, respectively. For *S. aureus*, the inhibition zones were 19.2 and 13.4 mm, respectively. *E. coli* showed a sensitivity to amoxicillin at 25 μg and cefoxitin at 30 μg, with

inhibition zones of 19.1 and 9.19 mm, respectively, and a resistance against ampicillin at 10 μg and cefadroxil at 30 μg. Ampicillin at 10 μg, amoxicillin at 25 μg, cefadroxil at 30 μg, cefoxitin at 30 μg, and vancomycin at 30 μg exhibited antibacterial effects against *S. aureus*, with values of 7.37, 21.2, 10.2, 11.2, and 11.2 mm, respectively, whilst *P. aeruginosa* were resistant towards all the aforementioned antibiotics.

**Table 2.** In vitro antibacterial activity of some selected standard antibiotics against the tested pathogenic bacterial strains using agar disc diffusion assay.

| Antibiotics | Average Diameters of Inhibition Zones (mm) | | |
|---|---|---|---|
| | *E. coli* | *P. aeruginosa* | *S. aureus* |
| ampicillin (10 μg) | – | – | 7.37 ± 0.31 [a] |
| amoxicillin (25 μg) | 19.1 ± 0.33 [b] | – | 21.2 ± 0.26 [a] |
| cefadroxil (30 μg) | – | – | 10.2 ± 0.18 [a] |
| doxycycline (30 μg) | 15.2 ± 0.22 [b] | 9.11 ± 0.08 [c] | 19.2 ± 0.22 [a] |
| cefoxitin (30 μg) | 9.19 ± 0.19 [b] | – | 11.2 ± 0.19 [a] |
| ofloxacin (5 μg) | 16.4 ± 0.36 [b] | 6.12 ± 0.0 [c] | 13.4 ± 0.29 [a] |
| vancomycin (30 μg) | – | – | 11.2 ± 0.22 [a] |

Notes: Data are represented as means ± SD ($n$ = 3). Means in the same row with different letters are significantly different at $p < 0.001$.

### 3.3. Minimum Inhibitory Concentrations (MICs) of R. hieroglyphicum Extracts

The minimum inhibitory concentrations (MICs) of the crude ethanolic, methanolic, and acetone extracts of the *R. hieroglyphicum* specimens are shown in Figure 1. In general, there was a potent antibacterial action from all the algal extracts towards all the tested bacterial strains, particularly the ethanolic extract, where its MICs values were 2.25, 0.533, and 5.34 mg mL$^{-1}$ against *E. coli*, *P. aeruginosa*, and *S. aureus*, respectively.

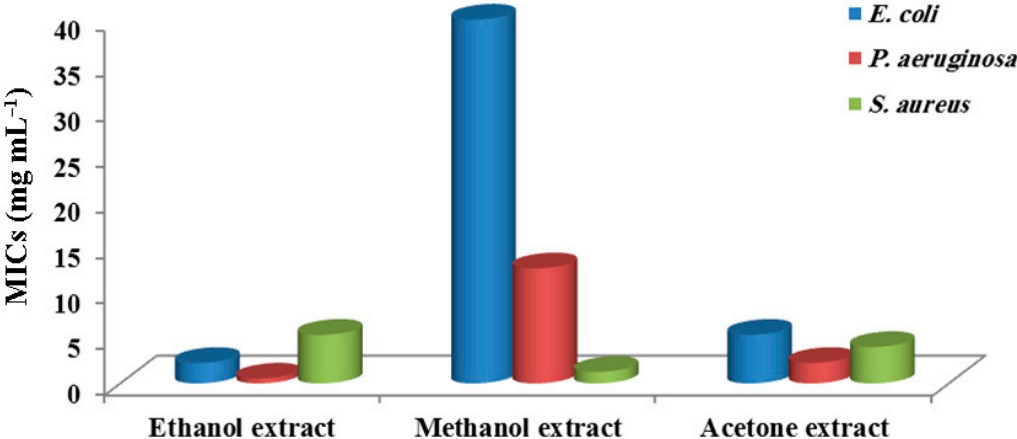

**Figure 1.** MICs of the different *Rhizoclonium hieroglyphicum* extracts against the three investigated pathogenic bacterial strains.

### 3.4. Phytochemical Constituents of the Rhizoclonium hieroglyphicum Ethanolic Extract

A total of 30 different bioactive compounds were identified in the ethanolic extract of *R. hieroglyphicum*. Saturated and polyunsaturated FAs (stearic, myristic, palmitic, α-linolenic, and oleic acids) constituted the main compounds, in addition to minor concentrations of ester derivatives, long chains of hydrocarbons, amide-containing compounds, and aldehydes (Table 3 and Figure 2).

Table 3. GC–MS analysis of the crude ethanolic extract of *Rhizoclonium hieroglyphicum* *.

| No. | Compounds Identified | RT (min) | Peak Area (%) | Norm (%) |
|---|---|---|---|---|
| 1 | 2-[2-[2-[2-[2-[2-(2-methoxyethoxy)ethoxy]ethoxy]ethoxy]ethoxy]ethoxy-trimethylsilane | 32.214 | 9.272 | 100.00 |
| 2 | Octadecanoic acid, ethyl ester (**stearic acid C18:0**) | 30.644 | 8.030 | 86.60 |
| 3 | 2-[2-[2-[2-[2-[2-[2-(2methoxyethoxy)ethoxy]ethoxy]ethoxy]ethoxy]ethoxy]ethoxy-trimethylsilane | 34.375 | 5.720 | 61.69 |
| 4 | tert-butyl-[2-[2-[2-[2-[2-[2-[2-[2-(2methoxyethoxy)ethoxy]ethoxy]ethoxy]ethoxy]ethoxy]ethoxy]ethoxy]dimethylsilane | 29.088 | 5.318 | 57.36 |
| 5 | Hexadecanoic acid, ethyl ester (**palmitic acid C16:0**) | 27.698 | 5.161 | 55.66 |
| 6 | {2,2-dimethyl-5-[2-(2-trimethylsilylethoxymethoxy)propyl][1,3]dioxolan-4-yl}methanol | 32.059 | 4.532 | 48.88 |
| 7 | 9,12,15-octadecatrienoic acid, 2-[(trimethylsilyl)oxy]-1-[[(trimethylsilyl)oxy]methyl]ethyl ester, (Z,Z,Z)-(**α-linolenic acid C18:3; ω−3**) | 34.540 | 4.427 | 47.75 |
| 8 | 1,4,7,10,13,16,19-heptaoxa-2-cycloheneicosanone | 28.988 | 3.827 | 41.28 |
| 9 | (1S,14S)-bicyclo[12.10.0]-3,6,9,12,15,18,21,24-octaoxatetracosane | 34.495 | 2.927 | 31.57 |
| 10 | **Oleic acid**, ethyl ester (**C18:1, ω−9**) | 36.456 | 2.752 | 29.68 |
| 11 | 1,2-hexanediol, 2-methyl- | 23.246 | 2.702 | 29.14 |
| 12 | 1,2-benzenedicarboxylic acid, diisooctyl ester | 34.020 | 1.578 | 17.02 |
| 13 | Cyclotetrasiloxane, octamethyl- | 8.405 | 1.473 | 15.89 |
| 14 | Tetradecanoic acid, ethyl ester (**myristic acid C14:0**) | 22.881 | 0.823 | 8.88 |
| 15 | 4-ethylbenzoic acid, 2-butyl ester | 7.014 | 0.778 | 8.39 |
| 16 | 2-dimethylsilyloxytridecane | 11.071 | 0.775 | 8.36 |
| 17 | [1,1′-bicyclopropyl]-2-octanoic acid, 2′-hexyl-, methyl ester | 32.549 | 0.721 | 7.78 |
| 18 | Cyclopentasiloxane, decamethyl | 11.706 | 0.629 | 6.78 |
| 19 | 3,7,11,15-tetramethyl-2-hexadecen-1-ol (phytol) | 29.548 | 0.616 | 6.65 |
| 20 | 1-propanol, 3,3′-oxybis- | 10.260 | 0.530 | 5.71 |
| 21 | 3-deoxyglucose | 8.765 | 0.435 | 4.69 |
| 22 | Cyclohexasiloxane, dodecamethyl- | 14.847 | 0.428 | 4.61 |
| 23 | 3-Isopropoxy-1,1,1,7,7,7-hexamethyl-3,5,5-tris(trimethylsiloxy)tetrasiloxane | 17.573 | 0.325 | 3.51 |
| 24 | Phenol, 2,2′-methylenebis[6-(1,1-dimethylethyl)-4-methyl- | 32.945 | 0.321 | 3.46 |
| 25 | 1,3,5-pentanetriol, 3-methyl | 10.185 | 0.301 | 3.24 |
| 26 | Bicyclo[2.1.1]hexan-2-ol, 2-ethenyl- | 6.419 | 0.276 | 2.98 |
| 27 | Octaethylene glycol monododecyl ether | 15.212 | 0.245 | 2.64 |
| 28 | Octasiloxane, 1,1,3,3,5,5,7,7,9,9,11,11,13,13,15,15-hexadecamethyl- | 35.541 | 0.242 | 2.61 |
| 29 | Cyclononasiloxane, octadecamethyl | 30.759 | 0.228 | 2.46 |
| 30 | Ethyl oleate | 30.294 | 0.225 | 2.43 |

Notes: * Peak areas (or percentage composition) of compounds identified are relative to each other within the same extract.

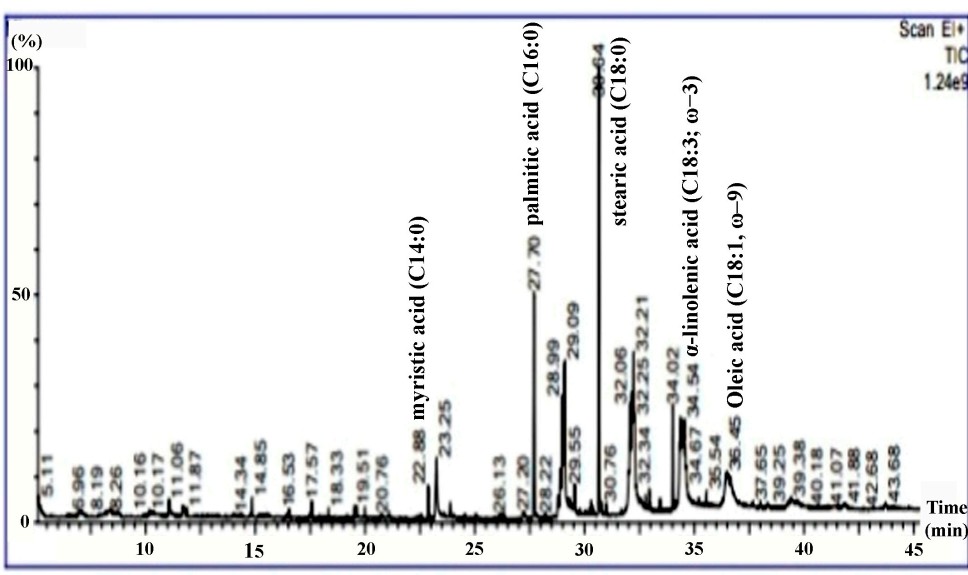

**Figure 2.** GC–MS chromatograph of the bioactive constituents (as % relative areas) of *Rhizoclonium hieroglyphicum* ethanolic extract investigated in this study.

## 4. Discussion

The discovery of novel and potent antimicrobial compounds of natural origins, such as algae and seaweeds, is currently the main task of many researchers in order to face the unexpected pathogenic resistance of some bacterial strains [9,13]. Algae-derived bioactive compounds generally offer a broad clinical effectiveness against a wide niche of human-infecting pathogens such as bacteria, fungi, and viruses [6,29,30]. Based on our findings, *R. hieroglyphicum* extracts possess significant antibacterial activity against all the investigated pathogenic bacteria, *E. coli*, *P. aeruginosa*, and *S. aureus*, in a concentration-dependent manner. Specifically, the polar ethanolic extract exhibited the most potent antibacterial activity. This antimicrobial activity is most likely attributed to the synergistic effects of the high-value bioactive compounds extracted by ethanol, particularly FAs. In agreement with our observations, Rani et al. [31] investigated the antibacterial activity of different polar and nonpolar solvents for Indian *R. hieroglyphicum* against *E. coli*, *S. aureus*, *Salmonella enterica* serovar *typhi*, and *Streptococcus mutans*, but they did not characterize its bioactive antibacterial compounds. However, they reported that the polar solvent "methanol" exhibited the highest antibacterial effect. Contrarily, Mungmai et al. [6] pinpointed that the ethanolic and aqueous extracts of *R. hieroglyphicum* specimens, isolated from the Nan River in northern Thailand, had no antimicrobial effect against methicillin-resistant *S. aureus*, *S. aureus* ATCC 29213, and *Propionibacterium acne* ATCC 6919 using an agar well diffusion method. In a similar study on microalgae, Morsi et al. [32] highlighted that *Scenedesmus obliquus* (today identified as *Tetradesmus obliquus*) extracts, particularly crude ethanolic ones, are distinguished by their powerful antibacterial activities against *E. coli*, *P. aeruginosa*, and *S. aureus*. They linked this antibacterial action to the presence of a combination of abundant bioactive constituents of unsaturated FAs and polyphenolics. Additionally, Yotova et al. [33] studied the FAs profile in the green coccid *Coelastrella* sp. BGV and identified 11 different FAs, within which, monounsaturated oleic acid (C18:1, $\omega$–9) was present in the largest amount at 31.7%, followed by polyunsaturated linoleic acid (C18:2, $\omega$–6; 25.9%) and saturated palmitic acid (C16:0; 19.2%). They reported that long-chain unsaturated FAs have varying degrees of antibacterial activity against both Gram-positive and Gram-negative pathogens. As a model on a motile green microalga, Sudalayandi et al. [34] attributed the potent antibacterial action of *Chlamydomonas reinhardtii* to its unsaturated FAs, including linolenic, oleic, and palmitoleic acids, as well as some saturated FAs such as myristic, palmitic, and stearic acids. All these bioactive antibacterial FAs have already been identified in Egyptian *R. hieroglyphicum*.

In the present study, the *R. hieroglyphicum* ethanolic extract was characterized by the occurrence of considerable amounts of FAs with variable saturation degrees (Table 3 and Figure 2). These findings highly coincide with the results obtained by Ghazala et al. [12] during their investigation on *R. hieroglyphicum* specimens isolated from freshwater channels in Pakistan. They recorded 16 different fatty acid species; most of them belonged to palmitic (C16:0), pentadecanoic (C15:0), caproleic (C10:1), margaric (C17:0), tridecanoic (C13:0), heptadecenoic acid (C17:1), oleic (C18:1), heneicosylic (C21:0), erucic (C22:1), behenic (C22:0), and carboceric (C27:0) acids. The synergistic action mainly exerted by saturated (mainly stearic, palmitic, and myristic), monounsaturated $\omega$–9 oleic, and polyunsaturated $\omega$–3 $\alpha$-linolenic FAs, as well as some other bioactive constituents, appeared to be responsible for the potent antibacterial activity of the Egyptian *R. hieroglyphicum* ethanolic extract (see Table 3). Regarding their mechanism of action, it is quite likely that the FAs changed the fluidity of the bacterial cell membrane and caused conformational changes in the membrane proteins, making the bacterial internal components seep out and ultimately causing cell death. Previous studies conducted by Pohl et al. [35] and Mohamed and Saber [7] give good support to our conclusion, in that they supposed that FAs are capable of inserting themselves into the lipid bilayer of microbial cell membranes, disrupting the electron transport chains and membranes, and ultimately causing increased fluidity and cellular damage. Another hypothesis for the antibacterial mechanism of FAs was proposed by Zheng et al. [36]. They examined whether unsaturated FAs are selective inhibitors

of FabI, an enoyl–acyl carrier protein reductase that plays an essential functional role in bacterial FA synthesis, using a $^{14}$C acetate incorporation assay. They found that linoleic acid, as a model fatty acid, could inhibit FabI; thus, it has been suggested as a potential target for antibacterial medications. They also pinpointed that the unsaturated oleic, palmitoleic, linolenic, and arachidonic acids exhibited the same FabI inhibition mechanism. In agreement with our study, both oleic and α-linolenic (C18:3; $\omega-3$) acids have been already present in the crude *R. hieroglyphicum* ethanolic extract. In more detail, long-chain unsaturated FAs could exhibit the inhibition of the FabI enzyme and therefore could stop the fatty acid production in bacterial membranes [36]. Confirming this hypothesis, they noticed that supplementation with either saturated FAs, such as stearic and palmitic, or unsaturated FAs, such as oleic, obviously reversed the antibacterial action of linoleic acid. This observation supports the hypothesis that unsaturated FAs could inhibit the fatty acid synthesis in bacterial cells and consequently have antibacterial effects. Additionally, it has been documented that long-chain unsaturated FAs are characterized by their potent bactericidal activity toward some pathogenic microorganisms, such as methicillin-resistant *S. aureus* [37], *Helicobacter pylori* [38], and *Mycobacteria* [39]. Supporting these previous findings, it has been reported that long-chain unsaturated FAs, including oleic, linoleic, and α-linolenic acids, are distinguished by these antibacterial activities, while long-chain saturated FAs, such as stearic and palmitic acids, are less active [38–41]. Therefore, the antibacterial action of the Egyptian *R. hieroglyphicum* ethanolic extract could be primarily linked to its bulk of long-chain unsaturated and saturated FAs.

Lastly, the Egyptian *R. hieroglyphicum* specimens could be cultivated and exploited in the large-scale industrial production of efficient and natural antibacterial products, aside from their valuable contents of functional and nutritional compounds. This will help us in beating the growing resistance of some pathogenic bacterial strains, and maybe other human pathogens, to commercial drugs.

## 5. Conclusions

The crude ethanolic, methanolic, and acetone extracts of the freshwater alga *Rhizoclonium hieroglyphicum*, mostly untapped in Egyptian inland waters, exhibited significant antibacterial activity against *E. coli*, *P. aeruginosa*, and *S. aureus*. However, the ethanolic extract was the most potent, and its antimicrobial action could be attributed to the synergistic action of its high-value bioactive fractions, particularly its long-chain polyunsaturated and saturated fatty acids. This upgraded antibacterial action, communicated in successive extraction, was most likely due to the way in which both the hydrophobic and hydrophilic bioactive constituents were separated. Accordingly, the *R. hieroglyphicum* ethanolic extract is highly recommended to be exploited for new antibacterial drug development. Nonetheless, further exploration of the antibacterial effect of each bioactive fraction is needed, in order to better understand the tested limits.

**Author Contributions:** Conceptualization, H.H.M., S.M.E.-S. and A.A.M.; methodology, A.D.M., H.H.M. and A.A.S.; software, A.D.M., H.H.M. and A.A.S.; validation, H.H.M., S.M.E.-S and A.A.M.; formal analysis, A.D.M.; investigation, A.D.M., S.M.E.-S, A.A.M. and H.H.M.; resources, A.D.M., H.H.M., S.M.E.-S and A.A.M.; data curation, H.H.M., S.M.E.-S., A.A.M., M.A.-H., A.E. and A.A.S.; writing—original draft preparation, A.D.M., H.H.M. and A.A.S.; writing—review and editing, H.H.M., A.A.S. and M.M.E.-S.; visualization, H.H.M., A.A.S. and M.M.E.-S.; supervision, H.H.M., S.M.E.-S., A.A.M. and M.M.E.-S.; project administration, H.H.M., S.M.E.-S. and A.A.M.; funding acquisition, H.H.M., S.M.E.-S., M.A.-H., A.E. and A.A.M. All authors have read and agreed to the published version of the manuscript.

**Funding:** This research received no external funding.

**Data Availability Statement:** Data are available upon request from the authors.

**Acknowledgments:** The authors are very grateful to the Department of Botany and Microbiology, Faculty of Science, Menoufia University, the Algae Unit at the Soil, Water and Environment Research Institute at Sakha Agriculture Research Station, Kafr El-Sheik, and the Botany Department, Faculty

of Science, Ain Shams University, Egypt for providing all facilities to perform this experimental work. The authors also thank the Princess Nourah bint Abdulrahman University Researchers Supporting Project number (PNURSP2023R182), Princess Nourah bint Abdulrahman University, Riyadh, Saudi Arabia.

**Conflicts of Interest:** The authors declare no conflict of interest.

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
