# Peer review of "Antibacterial Activity of Bioactive Compounds Extracted from the Egyptian Untapped Green Alga Rhizoclonium hieroglyphicum"

_water, doi:10.3390/w15112030_

Round 1

Reviewer 1 Report

Dear Authors, you have done good work, where you have tested the algal (Rhizoclonium hieroglyphicum) extract using ethanol, methanol, and acetone.

The pathogenic bacterial isolates you have used in this study- Staphylococcus aureus, Escherichia coli and Pseudomonas aeruginosa, did you have the data of antibiotic sensitivity biogram of these isolates?

Were you sure that polar solvent based algal extracts did  have only algal extract and not having solvent, since the antibacterial activity may be due the polar solvents.

You have mentioned a research paper (Ref. No. 6), how your work is different from this published study?

You have discussed M&M, Results, and discussion part nicely, try to incorporate the latest references 2020 onwards  (if available) with reference to the same bacteria you tested and some other algal genera.

Author Response

Reviewer #1:

  1. Dear Authors, you have done good work, where you have tested the algal (Rhizoclonium hieroglyphicum) extract using ethanol, methanol, and acetone. The pathogenic bacterial isolates you have used in this study - Staphylococcus aureus, Escherichia coli and Pseudomonas aeruginosa. Did you have the data of antibiotic sensitivity biogram of these isolates?.

R: Thank you for your valuable comments. The strains used were authentic strains with well-known antibiotic sensitivity biograms. In our study, as demonstrated in Table 2, we highlighted the antibiotic sensitivity effects of seven different antibiotics in terms of average diameters of inhibition zones.

  1. Were you sure that polar solvent based algal extracts did have only algal extract and not having solvent, since the antibacterial activity may be due the polar solvents.

R: At the beginning, the solvents were evaporated by a rotary evaporator (Büchi R-200, Marshall Scientific, Minnesota, USA) at 40°C., and then the crude algal extracts were re-dissolved in the respective solvent on application where each solvent was also tested alone to confirm the antibiotic algal effect.

  1. You have mentioned a research paper (Ref. No. 6), how your work is different from this published study?.

R: OK. Reference 6 (Mungmai et al. 2014) investigated the antibacterial effects of both the ethanolic and aqueous extracts of R. hieroglyphicum from northern Thailand against Staphylococcus aureus ATCC 29213, methicillin-resistant S. aureus and Propionibacterium acne ATCC 6919. These extracts had no effects against the test of microorganisms. Contrarily, we used three different polar extracts against different strains (Staphylococcus aureus, Escherichia coli and Pseudomonas aeruginosa), and noticeably they exhibited different niches of antibacterial activity and the R. hieroglyphicum ethanolic extract had the most potent antibacterial effect. Moreover, the study of Mungmai et al. (2014) only assessed the phenolic and polysaccharide contents of the R. hieroglyphicum, but our work characterized the bioactive compounds of the most potent solvent “the ethanolic extract” where 30 different bioactive constituents, mainly including long-chain polyunsaturated and saturated fatty acids, were identified.

  1. You have discussed M&M, Results, and Discussion part nicely, try to incorporate the latest references 2020 onwards (if available) with reference to the same bacteria you tested and some other algal genera.

R: Thank you. We added more related references.

Reviewer 2 Report

This manuscript describes a potentially interesting bioactivity for an extract.  However, the results are too preliminary and ambiguous to meet the standards required for the chemistry content.  Using crude extracts of microorganisms is full of pitfalls, and the normal standard would be to purify the active component(s) to chemical homogeneity using activity-directed fractionation,, and then demonstrate their bioactivities.  This has not been done in the present work.  Another major issue comes from the GC-MS analysis of the crude extract: while the fatty acids may well be natural, and responsible for the bioactivity, many of the identified components are clearly man-made and hence are contaminants.  These include for example all the silanes such as the first compound in Table 3, and the phthalate ester (line 253, a known plasticizer), yet there is no recognition that these strongly suggest contamination of the sample.  Some of these (e.g. Table 3 line 1) are present in large amounts, and could account for all the bioactivity, so it is not at all possible to conclude that the fatty acids are the active components.  This is why purification is so important.  A conclusion on the activity of the fatty acids could have been reached by simply testing the individual commercial acids, or even reconstituting the mixture in the correct proportion, yet this has not been performed.  So the statement that the proportions are important is not substantiated.

For these reasons, I cannot recommend publication of this work.

Author Response

Reviewer #2:

This manuscript describes a potentially interesting bioactivity for an extract.  However, the results are too preliminary and ambiguous to meet the standards required for the chemistry content.  Using crude extracts of microorganisms is full of pitfalls, and the normal standard would be to purify the active component(s) to chemical homogeneity using activity-directed fractionation, and then demonstrate their bioactivities.  This has not been done in the present work. Another major issue comes from the GC-MS analysis of the crude extract: while the fatty acids may well be natural, and responsible for the bioactivity, many of the identified components are clearly man-made and hence are contaminants. These include for example all the silanes such as the first compound in Table 3, and the phthalate ester (line 253, a known plasticizer), yet there is no recognition that these strongly suggest contamination of the sample.  Some of these (e.g. Table 3 line 1) are present in large amounts, and could account for all the bioactivity, so it is not at all possible to conclude that the fatty acids are the active components.  This is why purification is so important.  A conclusion on the activity of the fatty acids could have been reached by simply testing the individual commercial acids, or even reconstituting the mixture in the correct proportion, yet this has not been performed.  So the statement that the proportions are important is not substantiated.

For these reasons, I cannot recommend publication of this work.

R: Thank you for your comments. However, we would like to stress that we will continue on our preliminary data to assess the antibacterial effect of each major bioactive compound identified. Furthermore, due to characterizing 30 different constituents in the R. hieroglyphicum ethanolic extract, we attributed its potent antibacterial activity to the synergistic antibacterial activity of these compounds, particularly the major quantitative compounds “fatty acids”. 

Reviewer 3 Report

The topic of the article is very actual, as the discovery of alternative natural antibacterial medicines is more and more crucial nowadays. However, the description of how the experiments were carried out is inadequate and not reproducible. It is not proven that the presented results derive solely from the antimicrobial action of the Rhizoclonium glyphicum exctract and not derive from the solvents as well. The results cannot be accepted without checking the antimicrobial activity of the solvents (in the concentration used in the experiments) on their own.

Lines 31-32: Strain numbers (similarly to lines 65-66) must be indicated.

Line 50: Instead of C.Agardh - C. Agardh

Lines 108-109: Strain numbers (similarly to lines 65-66) must be indicated.

Line 116: It must be specified what percentage of ethanol, methanol and acetone was used for the extraction.

Lines 124-125: There is no such media that is called supplement agar meadia. Please define the name of the exact medium that was used for the experiment instead of supplement agar media. The mentioned paper plate assay is generally called agar disc diffusion method (as it was called later in Table 1) in antimicrobial assay literature. Please change it.

2.4. Antibacterial Assay of the Algal Extracts Did you scheck the antimicrobial effect of the solvents? When ethanol etc. are used as a solvent for an antimicrobial compound there antimicrobial effect shoud be checked as well. They might have inhibitory effect and therefore the results can originate from the combination of the antimicrobial coumpound and the solvent. To show the the effect that might be attributed only to the alagal extract these results also must be included in the study.

Line 137: What a supplement stock means? There is no such media. Please specify it and use exact culture broth names.

Line 138: instead of 2 x 106 - 2 x 106

At the end of 2.5. Determination of the Minimal Inhibitory Concentrations (MICs) of the Algal Extracts please also specify here how you determined the smallest dilution that did not have an antimicrobial effect.

Line 180: Please add the following sentence after the title of Table 1. Data represents the thickness of the inhibition zones in mm. Results of the solvents at the used concetrations must be presented as well.

Line 297: Salmonella Typhi is a serovar belonging to Salmonella enterica therefore the proper nomenclature is Salmonella enterica serovar Typhi or shortly Salmonella Typhi.

Author Response

Reviewer #3:

The topic of the article is very actual, as the discovery of alternative natural antibacterial medicines is more and more crucial nowadays. However, the description of how the experiments were carried out is inadequate and not reproducible. It is not proven that the presented results derive solely from the antimicrobial action of the Rhizoclonium hieroglyphicum extract and not derive from the solvents as well. The results cannot be accepted without checking the antimicrobial activity of the solvents (in the concentration used in the experiments) on their own.

R: Thank you. We edited it in the text.

Lines 31-32: Strain numbers (similarly to lines 65-66) must be indicated.

R: Done.

Line 50: Instead of C.Agardh - C. Agardh

R: Done.

Lines 108-109: Strain numbers (similarly to lines 65-66) must be indicated.

R: Done.

Line 116: It must be specified what percentage of ethanol, methanol and acetone was used for the extraction.

R: Edited.

Lines 124-125: There is no such media that is called supplement agar media. Please define the name of the exact medium that was used for the experiment instead of supplement agar media. The mentioned paper plate assay is generally called agar disc diffusion method (as it was called later in Table 1) in antimicrobial assay literature. Please change it.

R: Done. Thank you.

2.4. Antibacterial Assay of the Algal Extracts: Did you check the antimicrobial effect of the solvents? When ethanol etc. are used as a solvent for an antimicrobial compound there antimicrobial effect should be checked as well. They might have inhibitory effect and therefore the results can originate from the combination of the antimicrobial compound and the solvent. To show the effect that might be attributed only to the algal extract these results also must be included in the study.

R: We edited in the section Materials and Methods.

Line 137: What a supplement stock means? There is no such media. Please specify it and use exact culture broth names.

R: Done.

Line 138: instead of 2 x 106 - 2 x 106

R: Edited.

At the end of 2.5. Determination of the Minimal Inhibitory Concentrations (MICs) of the Algal Extracts please also specify here how you determined the smallest dilution that did not have an antimicrobial effect.

R: Done.

Line 180: Please add the following sentence after the title of Table 1. Data represents the thickness of the inhibition zones in mm. Results of the solvents at the used concentrations must be presented as well.

R: Edited.

Line 297: Salmonella Typhi is a serovar belonging to Salmonella enterica, therefore the proper nomenclature is Salmonella enterica serovar Typhi or shortly Salmonella tTyphi.

R: Edited. Thank you.

Reviewer 4 Report

In the manuscript provided by Saber and co-workers, antibacterial activity of the bioactive compounds extracted from the Egyptian untapped green alga Rhizoclonium hieroglyphicum has been assessed. In my opinion, although this article contains new aspects, the manuscript can be accepted with major revisions at Water.

- English writing needs further polish.

- The novelty is not sufficiently explained or clears in the introduction section. Also, the researches gap should be clearly described.

- Quality of the discussion section must be improved. In so doing, the authors must be organized the discussion from the general to the specific, linking your findings to the literature, then to theory, then practice and avoid repetition from the introduction.

- The "literature review" section of the manuscript is poor. It is necessary to compare the results of the present study with previous similar studies.

- Limitations of the study must be presented in the conclusion section.

- For numbers in text and tables < 1.00, use three digits beyond the decimal point; for numbers between 1.00 and 9.99 use two digits beyond the decimal point; for numbers between 10.0 and 99.9, use one digit beyond the decimal point; and for concentrations ≥ 100, use the nearest whole number.

Author Response

Reviewer #4:

In the manuscript provided by Saber and co-workers, antibacterial activity of the bioactive compounds extracted from the Egyptian untapped green alga Rhizoclonium hieroglyphicum has been assessed. In my opinion, although this article contains new aspects, the manuscript can be accepted with major revisions at Water.

- English writing needs further polish.

R: Done.

- The novelty is not sufficiently explained or clears in the Introduction section. Also, the research gap should be clearly described.

R: Edited.

- Quality of the Discussion section must be improved. In so doing, the authors must be organized the Discussion from the general to the specific, linking your findings to the literature, then to theory, then practice and avoid repetition from the Introduction.

R: We think it is well organized accordingly. Thank you.

- The “literature review” section of the manuscript is poor. It is necessary to compare the results of the present study with previous similar studies.

R: In the section Introduction we highlighted the antimicrobial effects of several cyanobacterial and algal species, and also stressed the role of several constituents including fatty acids. This coincides with our findings and we pinpointed it in the Discussion.  

- Limitations of the study must be presented in the Conclusion section.

R: Edited.

- For numbers in text and tables < 1.00, use three digits beyond the decimal point; for numbers between 1.00 and 9.99 use two digits beyond the decimal point; for numbers between 10.0 and 99.9, use one digit beyond the decimal point; and for concentrations ≥ 100, use the nearest whole number.

R: Edited. Thank you.

Reviewer 5 Report

I suggest the following changes and improvements:

1.     The first sentence of the abstract should be improved.

2.     The novelty of this work should be stated clearly in the introduction section.

3.     Provide a comparison-finding table of this study with other reported studies.

Author Response

Reviewer #5:

I suggest the following changes and improvements:

  1. The first sentence of the abstract should be improved.

R: We edited it.

  1. The novelty of this work should be stated clearly in the introduction section.

R: Done.

  1. Provide a comparison-finding table of this study with other reported studies.

R: Thank you for this note. Usually, the comparison table is provided to show the main differences and similarities of the key taxonomic features of new algal species. However, in the section Discussion we briefly discussed and compared our obtained results with the previous investigations and available literature: Mungmai et al. [6], Ghazala et al. [12], and Rani et al. [31].

Reviewer 6 Report

The article is well written. However, it must be clerified/improved for the given points:

1. How did you maintain photocycle? It is important to mention under the relevant heading.

2. Replace the heading “Origin of the bacterial strain” with “Bacterial strain”.

3. Add error bars in figures.

4. When bioactive compounds are mentioned in tables, then it is not necessary to present fig. 2.

5. Second-level discussion of the article is needed.

Author Response

Reviewer #6:

The article is well written. However, it must be clerified/improved for the given points:

  1. How did you maintain photocycle? It is important to mention under the relevant heading.

R: Edited.

  1. Replace the heading “Origin of the bacterial strain” with “Bacterial strain”.

R: Done.

  1. Add error bars in figures.
  2. OK. In Figure 1, where we assessed the MICs, this assessment is mainly based on the dilution technique. Briefly, the algal extracts were gradually diluted in test tubes in the range of 0.225 until 40 mg mL–1. 0.1 mL of each bacterial suspension, containing 2 x 106 CFU mL–1, was added in each tube. The tubes were incubated at the standard growth conditions (37 °C for 24 h). MIC was determined as the point at which there was 100% inhibition of the bacterial growth as compared to the control. This value is usually taken/recorded for one time and no need for replicates due to the very low concentrations. There are several published articles followed the same protocol, e.g.

- El-Sheekh MM, Daboor SM, Swelim MA, Mohamed S. Production and characterization of antimicrobial active substance from Spirulina platensis. 2014. Iran. J. Microbiol. 6(2): 112–9.

- Mohamed SS, Saber AA. 2019. Antifungal potential of the bioactive constituents in extracts of the mostly untapped brown seaweed Hormophysa cuneiformis from the Egyptian coastal waters. Egypt. J. Bot. 59(3): 695–708.

  1. When bioactive compounds are mentioned in tables, then it is not necessary to present fig. 2.

To be honest and also to be strong evidence for our valuable data, we think it would be much better to keep Fig. 2 which shows peaks of fatty acids identified from the Rhizoclonium hieroglyphicum ethanolic extract as the most powerful solvent.

  1. Second-level discussion of the article is needed.

R: Done.

Round 2

Reviewer 2 Report

While the authors have addressed some minor points, they have not dealt with the major issues I identified at the first review.  Table 3 still lists the many anthropogenic ingredients identified in the extract.  Indeed, the major components are fatty acids, but that doesn't mean they are the active components.  The authors have added solvent controls, but those do not control for the activity of these silicones and other non-natural components.  While the bioactivity may arise from the fatty acids, this is not proven and so this work does not meet rigorous scientific standards.  I cannot support publication until the authors have demonstrated which ingredient(s) is or are active.  This is needed to meet the normal standards of natural product research.

Author Response

R: Thank you. We thus linked the antibacterial action of the Egyptian R. hieroglyphicum to the synergistic effect of its crude ethanolic extract and demonstrated the mechanism to the major FAs.

Reviewer 4 Report

The revised manuscript has addressed my concerns and I agree with its publication after minor revision in Water.

-  The quality of the discussion section must be improved.

Author Response

R: Thank you for the respected reviewer. We have edited it.
